# Proactive Immunotherapeutic Approaches against Inflammatory Breast Cancer May Improve Patient Outcomes

**DOI:** 10.3390/cells11182850

**Published:** 2022-09-13

**Authors:** Daniel Alonso-Miguel, Steven Fiering, Hugo Arias-Pulido

**Affiliations:** 1Department of Animal Medicine and Surgery, Veterinary Medicine School, Complutense University of Madrid, 28040 Madrid, Spain; 2Department of Microbiology and Immunology, and Dartmouth Cancer Center, Geisel School of Medicine at Dartmouth and Dartmouth Hitchcock Health, Lebanon, NH 03756, USA

**Keywords:** inflammatory breast cancer, neoadjuvant immunotherapy, triple-negative breast cancer, immune checkpoint inhibitors, pembrolizumab, immunomodulators, in situ vaccination, cowpea mosaic virus nanoparticles, toll-like receptors

## Abstract

Inflammatory breast cancer (IBC) is highly metastatic at the onset of the disease with no IBC-specific treatments, resulting in dismal patient survival. IBC treatment is a clear unmet clinical need. This commentary highlights findings from a recent seminal approach in which pembrolizumab, a checkpoint inhibitor against programmed cell death protein 1 (PD-1), was provided to a triple-negative IBC patient as a neoadjuvant immune therapy combined with anthracycline–taxane-based chemotherapy. We highlight the findings of the case report and offer a perspective on taking a proactive approach to deploy approved immune checkpoint inhibitors. On the basis of our recently published research study, we propose in situ vaccination with direct injection of immunostimulatory agents into the tumor as an option to improve outcomes safely, effectively, and economically for IBC patients.

## 1. The Clinical Burden of IBC

Inflammatory breast cancer (IBC) is a rare, aggressive, and highly metastatic form of breast cancer (BC). IBC accounts for roughly 2.5% of all newly diagnosed BC in the US but is responsible for ~10% of BC-related deaths [1,2], making it the deadliest form of BC. Among the BC subtypes, triple-negative (TN) IBC accounts for 20% to 40% of all IBC cases, and within this subtype, ~30% present with lymph node and distant metastasis at diagnosis and have the poorest prognosis [3,4]. Anthracycline–taxane-based chemotherapy remains the backbone of neoadjuvant therapy for TN IBC; however, the efficacy of this treatment is poor, with a dismal 15-year survival rate of ~20–30% [5,6,7]. The lack of effective therapies against TN IBC is due, in part, to a lack of clinical trials because (1) TN IBC is a different clinical and molecular entity than TN non-IBC, which causes the exclusion of TN IBC patients from most clinical trials [8,9,10] and (2) IBC is rare, making patient recruitment difficult for specific TN IBC clinical trials.

Lacking specific treatments, TN IBC patients are relegated to the same treatments as noninflammatory TN BC patients. Furthermore, systemic targeted therapy of IBC has been extrapolated from studies on high-risk non-IBC patients, and drugs have been selected with little consideration for IBC biology or potential impacts on metastasis [11,12]. As a result, there have been minimal efforts and strategies to identify improved treatments for TN IBC patients [13]. This is highlighted by the fact that of the three clinical trials currently underway in the US evaluating immune checkpoint inhibitors (ICIs) in combination with chemotherapy or targeted therapy in IBC patients, there is only one trial involving recurrent or metastatic TN IBC patients (NCT03202316) (https://www.clinicaltrials.gov/ct2/results?cond=&term=NCT03202316&cntry=&state=&city=&dist= Accessed on 9 September 2022).

Hence, therapeutically, TN IBC is an orphan disease with a high unmet medical need.

Given the intrinsic aggressive and metastatic nature of IBC, an aggressive approach to tackle this deadly disease is needed to improve patient outcomes. In this light, the recent case report by Kharel et al. [14] demonstrating the use of neoadjuvant chemotherapy in combination with a T-cell checkpoint inhibitor (pembrolizumab, a PD-1 inhibitor) to treat a TN IBC patient offers a bright light at the end of the dark tunnel faced by TN IBC patients.

Kharel’s study reports the treatment of a TN IBC patient (Figure 1) who was given the KEYNOTE-522 clinical trial regimen: the patient received systemic intravenous neoadjuvant therapy comprising carboplatin (area under the curve 5 every 3 weeks), pembrolizumab (200 mg every 3 weeks), and weekly paclitaxel (80 mg/m^2^ for 12 weeks), followed by a combination of doxorubicin (60 mg/m^2^), cyclophosphamide (600 mg/m^2^), and pembrolizumab (200 mg every 3 weeks) for four cycles [14]. Although the phase 3 KEYNOTE-522 clinical trial was designed for previously untreated stage II or stage III TN breast cancer patients [15], including IBC, the last interim report on 602 patients who underwent randomization did not report any TN IBC patients [16]. Hence, the study by Kharel represents the first report of the efficacy of an ICI in the neoadjuvant setting against TN IBC and provides the following insights into the clinical relevance of this approach to treating TN IBC patients.

**A. Efficacious front-line agents:** The impressive clinical and pathological response observed in this patient can be related, among other factors, to the presence of high levels of tumor-infiltrating lymphocytes (TILs) and the combination of immunogenic cell death (ICD) inducers with an ICI (pembrolizumab) as front-line agents. Cancer cells undergo bona fide ICD in response to the ICD inducers (paclitaxel, doxorubicin, and cyclophosphamide) [17,18]. During ICD, dying cells release a panel of immunostimulatory damage-associated molecular patterns and cytokines that support the recruitment, phagocytic activity, and maturation of antigen-presenting cells, enabling them to engulf antigenic material, migrate to lymph nodes, and prime a cytotoxic T lymphocyte-dependent immune response [19]. Adding an anti-PD-1 inhibitor releases the immune T cells from immunosuppression through PD-1 and potentiates the ICD-generated systemic cytotoxic immune response. The establishment of immunological memory can translate into longer-term suppression of metastatic events and improved patient survival. It is expected, therefore, that this TN IBC patient will have a better outcome due to the treatment and compared with historical cases. This patient started chemo- and immunotherapy in August 2020 and underwent surgery in March 2021, followed by maintenance immunotherapy, which was completed in December 2021. As of this date, the patient is still clinically well and has no signs of recurrence [20].

**B. Biomarkers:** The importance of TILs and the protein levels of programmed death-ligand 1 (PD-L1) in IBC pathology and as surrogate biomarkers of patient outcomes has been demonstrated by us [21,22] and others [23,24]. The high TILs levels (~40%), PD-L1 positivity in a pre-treatment biopsy, and the excellent response in this TN IBC patient confirm the value of TILs and PD-L1 status in TN IBC patients [21,22,23,24]. It should be noted that while the efficacy of ICIs has been associated with PD-L1 positivity in metastatic TN BC patients [25,26], no association with PD-L1 protein levels was found in the KEYNOTE-522 trial [16]. The difference in results may be related to the different drugs or inhibition pathways, disease stages (early rather than late), PD-L1 assays, or all these factors [16]. Correlative genomic studies to identify molecular and immune biomarkers associated with TN BC patients’ responses to the pembrolizumab and chemotherapy regimen in the KEYNOTE-522 trial are planned. A transcriptomic analysis will provide a view of immune and cellular markers associated with the response to checkpoint inhibition [16].

**C. Pembrolizumab–chemotherapy tolerance:** The pembrolizumab–chemotherapy treatment was well tolerated, with only alopecia and grade 1 fatigue reported in this TN IBC patient; similar side effects were observed in the KEYNOTE-522 study [16]. It should be noted that although severe adverse events were reported in ~33% of the patients in the pembrolizumab–chemotherapy group in the KEYNOTE-522 study [16], the observed severe adverse events did not negatively affect the ability to administer the neoadjuvant chemotherapy. This is important since the administration of fewer doses of neoadjuvant chemotherapy than planned has been associated with worse long-term outcomes [27]. The good response with minimum toxicity in the TN IBC patient may be related, among other factors, to the presence of a responsive immune system, undamaged by previous grueling doses of toxic chemotherapy regimens as observed in heavily treated metastatic breast cancer patients undergoing immune checkpoint therapy [25].

## 2. The Urgency of Applying Novel Efficacious and Economically Available Therapies against IBC

IBC is highly metastatic, and despite deploying a multidisciplinary approach to tackle this disease, the current anthracycline–taxane approach is minimally effective, with a very low 15-year survival rate of ~20–30% [5,6,7]. In other words, IBC patients do not have the luxury of waiting until efficacious therapies are found.

While the biological toxicity was manageable in the neoadjuvant setting in Kharel’s report and the KEYNOTE-522 study [14,16], the economic toxicity associated with the very high cost of immunotherapies in cancer patients creates a huge economic personal and societal burden [28]. Hence, a new aggressive approach is needed to make efficacious immunotherapies available to patients without causing an economic burden.

Intratumoral administration of immunotherapeutic agents (in situ vaccination, ISV) is an attractive way to stimulate antitumor immunity while reducing both systemic and economic toxicity. Every vaccine contains an antigen and an immune adjuvant. The antigen is what the immune system is being trained to recognize and respond to, and the immune adjuvant stimulates the immune system to respond to the antigen(s). There are a variety of classes of tumor antigens: tumor-associated antigens, neoantigens, and, in some tumors, viral antigens [29]. All are within the tumor itself, but the immune response against these antigens is generally weak because the tumor creates an immunosuppressive local environment that suppresses a robust antitumor immune response. ISV uses the tumor as the source of the relevant antigen and introduces some form of immune stimulation (immune adjuvant) into the tumor to disrupt the local immune suppression and prime a new response or enhance a pre-existing antitumor immune response. This stimulates a local antitumor immune response, and more importantly, generates a systemic immune response against any tumor antigens shared by most or all metastatic and micro-metastatic foci in the cancer patient [30,31]. Clinically, this might translate into tumor reduction in the injected lesion; an expanded systemic immune response that suppresses noninjected tumors; and a significant delay or blocking of metastatic events, which are the main cause of cancer deaths. Of note, intratumoral treatment enables a high local concentration of immune stimulatory reagents while the systemic levels are low, which means that adverse immune-mediated events are reduced. The widely used checkpoint blockade therapies, such as PD-1- or PD-L1-blocking antibodies, depend on freeing tumor-recognizing effector T cells from the suppression of T-cell checkpoint molecules. Checkpoint blockade is most effective when there are many tumor-recognizing T cells. ISV can synergize with checkpoint blockade because it generates tumor-recognizing effector T cells and thus creates more of the proper T cells for checkpoint blockade therapy to work with [32,33].

The interest in ISV is reflected in the increasing number of clinical trials deploying various intratumoral immune stimulatory agents in a wide range of tumors [32,34]. While a review of ISV is not within the scope of this commentary, the scientific rationale and technical and clinical details of ISV can be found in recently published updates on this topic [32,33,34,35]. Relevant to this commentary, we recently demonstrated that ISV is indeed a potential novel and efficacious approach against IBC disease by demonstrating that ISV with empty cowpea mosaic virus (eCPMV) nanoparticles resulted in impressive clinical responses in five canine inflammatory mammary cancer (IMC) patients [36].

The preclinical efficacy of ISV using the highly immunostimulatory CPMV and eCPMV nanoparticles has been previously and extensively documented in various syngeneic murine tumor models, including a murine breast cancer model [37,38,39,40]. The excellent efficacy observed for CPMV/eCPMV immunotherapy in murine models was extended to more ‘translational’ models to facilitate the rapid deployment of CPMV-based therapy in humans by evaluating eCPMV ISV in companion dogs diagnosed with oral melanomas [41] and canine IMC patients [36]. IMC is the canine counterpart of human IBC, with similar clinicopathologic and biologic features and poor outcomes [42]. As such, canine IMC represents the optimal animal model to clinically translate efficacious therapies to humans while at the same time providing canine patients with state-of-the-art immunotherapies not yet available to humans [42].

Similar to the striking clinical and pathological response observed in the reported TN IBC patient treated with systemic neoadjuvant anti-PD-1 therapy [14], we also observed a robust clinical efficacy of neoadjuvant ISV eCPMV immunotherapy leading to tumor reduction in all five treated IMC dogs. Two of the IMC canine patients had such a reduction in tumor burden that they underwent surgery. Of note, surgery is not recommended for canine IMC patients because of the extensive local involvement, the presence of coagulopathies, and distant metastatic disease [43]. To our knowledge, this is the first report of an immunotherapy-based approach that allowed surgical intervention in IMC patients. Furthermore, no adverse events were seen in any of the eCPMV-treated dogs, and the quality of life was improved in three canine IMC patients. While canine IMC patients usually die of the disease within an average of one month without treatment [44], eCMPV therapy was associated with statistically significant improvement in the survival of treated dogs, and one IMC patient remained alive for up to ~6 months; the dog died of renal failure with no evidence of metastatic disease in the kidneys.

ICIs target specific molecules, and their efficacy is associated with the presence of the target molecule. CPMV is a plant virus and does not infect mammalian cells, meaning that it is not an oncolytic virus. CPMV nanoparticles are ‘tumor agnostic’; their efficacy is associated with the immune modulation of the tumor microenvironment; and they do not have a direct effect on tumor cancer cells. CPMV and eCPMV are identical in their protein content, but eCPMV lacks RNA, so it cannot infect anything. CPMV nanoparticles are agonists for toll-like receptors (TLRs) 2, 4, and 7, with TLR 2 and 4 recognizing the viral capsid and TLR7 recognizing the viral RNA genome. Because eCPMV lacks RNA, it exerts activity only through TLR2 and TLR4 [45].

Like the human TN IBC patient treated with neoadjuvant pembrolizumab and chemotherapy [14], all five canine IMC patients treated with eCPMV immunotherapy also had the TN tumor subtype. These were large tumors (the largest diameter of the treated tumors ranged from 4 cm to 20 cm). Although targeting the PD-1/PD-L1 axis has been linked mostly to PD-L1-positive TN patients, clinical responses to PD-1/PD-L1-blocking antibodies are not observed in all TN patients [25]. Of note, all of our TN IMC patients responded to eCPMV ISV therapy, and similar striking responses to eCPMV therapy have been observed in non-IMC hormone receptor-positive canine mammary cancer patients without a need for a specific marker [46]. The extensive data obtained using murine models [37,38,39,40], canine oral melanoma patients [41], and canine IMC patients [36] suggest that ISV with CPMV nanoparticles could be an effective immunotherapy for other solid tumors. Adding other immune checkpoint and/or immunomodulators will enhance the observed clinical efficacy of ISV CPMV immunotherapy [47].

## 3. Can We Achieve Good Clinical Responses and Improve Outcomes with Low Biologic and Economic Toxicity for IBC Patients?

While the striking positive response to neoadjuvant pembrolizumab and chemotherapy in the TN IBC patient provides a strong motivation to deploy existing FDA-approved systemic immunotherapies as front-line therapies against IBC disease to improve outcomes in IBC patients, the question of cost is still a huge burden for cancer patients, including IBC patients, because insurance companies may deny coverage. To provide a perspective on the high cost of immunotherapies, ipilimumab (an anti-CTLA-4 antibody) systemic therapy consists of 10 mg/kg every three weeks for four doses, followed by every three months for up to three years or until recurrence or unacceptable toxicity occurred. It has been estimated that the adjuvant ipilimumab regimen for melanoma has a price tag of USD 1.8 million per patient [48]. While no estimates exist yet for the cost of pembrolizumab or atezolizumab for breast cancer, it should be noted that the current systemic dose of pembrolizumab is 200 mg every three weeks for up to 24 weeks [14,16] and the dose of atezolizumab is 840 mg on day one and day 15 for ~24 weeks [25]. Clearly, checkpoint blockade therapy is not economically accessible for many cancer patients, including IBC patients.

From this economic standpoint, ISV therapy is very attractive for the simple reason that it requires a smaller amount of the immunotherapeutic agent, implying a lower cost without affecting the efficacy and minimal or no toxicity, which often limits the systemic counterpart. For example, ISV applying ipilimumab (just 2 mg once a week for 8 weeks) and interleukin-2 (IL-2) generated responses in both injected and noninjected lesions, with minimal additional toxicity in advanced melanoma [49]. This melanoma study supports the feasibility of ISV with ICIs as a therapeutic approach with excellent local and systemic (abscopal effect) efficacy and a significantly lower cost. It should be noted that most of the current ongoing clinical trials for breast cancer will apply ISV therapy with oncolytic viruses, immunomodulators, and cellular therapies in combination with chemotherapy and systemic immune checkpoint blockade [50]. Hence, efficacy will be observed in some of those trials, but the prohibitive cost of immune checkpoint blockade remains a major issue. ISV immunotherapy, therefore, represents a potential option to achieve striking clinical responses with minimal biological and economic toxicity. The pros and cons of ISV have been described in a few recent reviews [32,33,51]. In relation to IBC disease, the lack of solid masses in IBC may represent a technical challenge for ISV [8,10]. However, advances in image-guided techniques for intratumoral immunotherapy delivery will facilitate the application of ISV in IBC patients [52].

## 4. What Other Immunotherapies Could Be Applied with ISV to IBC Patients?

The intrinsic aggressive and metastatic nature of IBC leaves no room for clinical trials in which patients will probably die before ending their participation in the trial. We envision that a proactive approach to alleviate the burden suffered by IBC patients should utilize ISV on three fronts: 

1. Immunotherapies with existing and approved ICIs as front-line therapies. Kharel’s single case report of systemic therapy with good efficacy supports the use of ICIs as front-line therapies [14], and the melanoma study demonstrated the feasibility and high efficacy of ISV of a checkpoint inhibitor with good responses in the injected and noninjected advanced melanoma tumors [49].

2. Immunomodulators with excellent efficacy in preclinical models, giving preference to agents demonstrating high preclinical efficacy in mouse and canine models and low toxicity of ISV in humans and optimal models, such as dogs. The observed striking responses in canine IMC patients support studies to treat IBC patients with neoadjuvant ISV CPMV alone or in combination with standard-of-care therapy or other ICIs [36]. The current list of immunomodulatory agents being tested with ISV alone or in combination with systemic chemotherapy and checkpoint blockade in human breast cancer was detailed in Huppert’s review [50]. Not included in that review were a few recent reports of ISV immunomodulatory agents demonstrating impressive responses in injected and noninjected tumors in human patients with solid tumors [53,54] and canine cancer patients [54,55]. These compounds, in our view, represent potential immunotherapeutic options for IBC patients.

The Milhem group reported an update of an ongoing open-label, multicenter phase 1b/2 study (NCT03684785) of ISV using cavrotolimod, a TLR9 agonist, in combination with intravenous pembrolizumab in patients with large advanced solid tumors, including melanomas, Merkel cell carcinomas, cutaneous squamous cell carcinomas, head and neck squamous cell carcinoma, leiomyosarcoma, and various metastatic tumor patients [53]. Multiplex immunohistochemistry demonstrated an increase in CD8^+^ T cells and CD45RO^+^ memory T cells in the injected tumor lesion of a responder patient after treatment with cavrotolimod and pembrolizumab. Along with tumor reduction in the treated tumor, systemic immune activation was observed in noninjected tumor lesions distant from the site of the injection. The therapy was well-tolerated, with flu-like symptoms being the most severe adverse event reported. Durable and ongoing responses in all responders were observed, which is obviously quite impressive and well above current immunotherapy expectations [53].

The Fahrer group demonstrated the clinical efficacy of intratumoral injections of a slow-release emulsion of Complete Freund’s Adjuvant (CFA) containing heat-killed *Mycobacteria* in three preclinical species (relatively large mouse mastocytomas, canine mastocytomas, and equine melanomas) and in large human cancer patients (one nonsmall cell lung cancer, one metastatic osteosarcoma, one squamous cell cervical carcinoma, one squamous cell head and neck cancer, one prostate cancer, two metastatic renal cancers, two lung cancers, one urothelial cancer, and two invasive hormone receptor-positive ductal breast cancers) [54]. ISV of CFA was safe and well-tolerated in human, equine, and canine patients, with minor adverse events, such as inflammation at the site of injection and fever. As expected, a systemic immune response was also observed with regression of noninjected metastases. Of interest, analysis of immune cells infiltrating mastocytomas in mice showed that early neutrophil infiltration was predictive of treatment benefit; in addition, the regression of treated canine mastocytomas weeks or months after treatment demonstrated an increase in B- and T-cell infiltrates. The findings of this study suggest that activation of the innate immune system alone may be sufficient for the regression of some injected tumors, followed by activation of the adaptive immune system, which can mediate the regression of noninjected metastases [54].

This study is remarkable not only for the safety and efficacy of the ISV approach used here but also for the low cost of this CFA-based immunotherapy. CFA is licensed for human use, and dead *Mycobacteria* are inherently safer than the live *Mycobacteria* used both in the widely administered childhood intradermal bacille Calmett–-Guerin (BCG) vaccine against tuberculosis and as an intravesical treatment for superficial bladder cancer. As pointed out in Fahrer’s study, if correctly emulsified, CFA forms a slow-release depot at the injection site, meaning that one injection provides continuous immunostimulation over a period of weeks [54]. In relation to the cost of the CFA-based therapy, the ISV CFA injections in human patients consist of one to four injections with a volume ranging from 0.5 mL to 2 mL. The cost of the CFA used in Fahrer’s study was ~USD 38 for a 10 mL vial (each mL contained 1 mg of Mycobacterium tuberculosis (H 37RA), heat-killed and dried, 0.85 mL of paraffin oil, and 0.15 mL of mannide monooleate).

This immunotherapeutic approach could be very attractive for IBC patients in North African countries; the IBC incidence ranges from ~5% to 11% of total breast cancers in Morocco, Algeria, Tunisia, and Egypt [22,56,57,58], making this specific disease a high economic burden for these countries. Hence, it is hard to disagree with Fahrer’s conclusion that “CFA therefore has the potential to be a simple and inexpensive form of cancer immunotherapy, accessible for use in both developed and emerging economies” [54]. It is our hope that given CFA’s fewer side effects, good efficacy, and low cost, this CFA-based study will attract the attention of IBC oncologists to deploy this therapy in large-scale studies in those countries where the issue of enrolling IBC patients is not a hurdle as it is in other countries where IBC is rare and enrolling an appropriate number of patients is a huge problem [1,2].

Although IL-2 and IL-12, which expand and stimulate T cells and natural killer (NK) cells to mediate antitumor immunity, have demonstrated promising therapeutic effects, their clinical use is limited by severe systemic adverse effects [59]. The Wittrup group has developed IL-2 and IL-12 cytokine fusion proteins anchored to collagen, which is abundantly and ubiquitously expressed in tumors, and tagged the cytokines with canine serum albumin to increase the molecular weight of the proteins in order to prolong their intratumoral retention and effectively eliminate toxic systemic exposure [60,61]. This smart design demonstrated high activity and tolerability in large canine soft tissue sarcoma (STS) and canine oral melanoma (OM) patients [55]. One to two ISV injections with IL-2 and IL-12 fusion proteins were performed as neoadjuvant therapy in 10 canine STS cases, followed by tumor surgery. Twelve canine OM cases received the ISV injection in combination with radiation therapy the same day, followed by five ISV injections of IL-2/IL-12 fusion proteins every two weeks and follow-up of OM patients. The ISV IL-2/IL-12 therapy was well-tolerated, with transient body temperature elevation as well as mild neutropenia/thrombocytopenia. Evidence of a systemic response was observed in one OM patient by a significant tumor reduction in a lung metastatic site. IHC analysis of tissues from treated STS canine patients demonstrated enhanced infiltration, and RNA profiling reveal enrichment of genes associated with antitumor effector function. In addition, IL-2/IL-12 treatment in canine STS increased CTLA-4 and PD-L1 expression, implying that adding ICIs could potentiate T- and NK-cell effector functions [55]. These findings support the safety and efficacy of ISV with IL-2/IL-12 collagen binding cytokines alone or combined with standard-of-care surgery, radiation therapy, and other ICIs.

3. Treating a heterogeneous disease with complex combination therapies. Although the ISV immunotherapies described above have demonstrated striking clinical responses, we should remember that, as observed with the current ICIs approved to treat many cancers, these agents do not benefit all patients. Relapses are frequent because redundant pathways of immune suppression are active, important pathways of immune activation are silent, and critical tumor signaling pathways remain intact [62]. Hence, to overcome resistance and extend the benefit of immunotherapies to more patients, we need to precisely modulate various immunoregulatory and tumor intrinsic pathways. A series of safe and affordable agents are approved or in clinical trials and demonstrate good efficacy, including known chemotherapy drugs (doxorubicin, paclitaxel, and cyclophosphamide), small inhibitor molecules (AKT inhibitors, MEK inhibitors, and PARP inhibitors), and TLR agonists (SD-101, an agonist for TLR9). Some of these agents are deployed as dual therapies in various cancers. The details of the clinical trials in BC of these agents can be found in recent detailed reviews [50,62,63,64]. Furthermore, the ISV approach requires small quantities injected into the tumor parenchyma, reducing cost and potentially toxic events as well as offering the unique opportunity to combine multiple therapeutic agents to achieve a more robust immune response, higher objective response rates, and increased response duration in cancer patients, including IBC patients. While it is not the focus of this commentary, overcoming intrinsic resistance to therapies other than immunotherapy also requires ongoing and consistent consideration of new rational drug combinations that would be safe, affordable, and effective.

## 5. Conclusions

There is sufficient data to support the application of ISV in IBC patients using the currently approved ICIs alone or in combination with standard of care in the neoadjuvant setting to take advantage of a chemo-naïve immune system not yet affected by toxicity from established chemotherapeutic regimens. Adding new and potent immunomodulators, such as CPMV, cavrotolimod, CFA, and IL-2/IL-12, and other safe, affordable, and potentially effective therapies against this extremely aggressive and metastatic disease provides a pathway to significantly improve patients’ outcomes and eliminate IBC as a therapeutic orphan disease.

## Figures and Tables

**Figure 1 cells-11-02850-f001:**
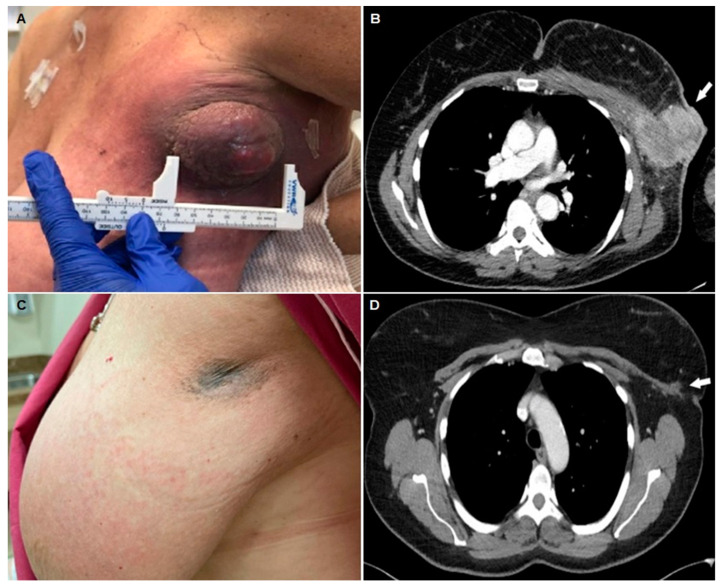
**Neoadjuvant pembrolizumab and chemotherapy induced clinical response in a triple-negative IBC patient.** A small boil-like left axillary mass increased in size in six weeks with associated pain, swelling, and erythema (**A**). Chest computerized tomography (CT) indicated the presence of a large left fungating mass in the lateral left breast (7.7 × 5.7 cm) with associated left sub-pectoral, axillary, and internal mammary lymphadenopathy and diffuse edema and skin thickening within the left breast with skin involvement (white arrow) (**B**). The fungating mass and left breast erythema were resolved after completion of the neoadjuvant pembrolizumab and chemotherapy (**C**); a post-treatment chest CT with contrast demonstrated a significant decrease in the size of the previously biopsied left axillary mass (white arrow) (**D**). Improvement in previously seen left axillary lymphadenopathy and dermal thickening of the left breast/axilla was also observed (**D**). Text and pictures (**A**–**D**) were modified from Kharel et al. [14], Breast Disease (2022); 41:255–260 (http://dx.doi.org/10.3233/BD-210041 Accessed on 9 September 2022).

## Data Availability

Not applicable.

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
