# Peer review of "Proactive Immunotherapeutic Approaches against Inflammatory Breast Cancer May Improve Patient Outcomes"

_cells, 2022, doi:10.3390/cells11182850_

Round 1
Reviewer 1 Report
In this commentary, Alonso-Miguel et al. described proactive immunotherapy approaches for improving outcomes in inflammatory breast cancer, which is relative rare but very aggressive subtype. They did a good job in describing the limitations of current approaches in dealing with aggressive breast cancers such as IBC. They discussed the issues pertaining to clinical trials in sufficient detail and provided a good rationale for proactive immunotherapies that would be effective and financially affordable. Overall, the commentary is well written and suitable for publication in Cells.
I have a few minor suggestions for the authors to consider.
1) Authors suggested that based on the criteria of potential efficacy and low cost, some of the proposed therapies may be more suitable for clinical trials in poor countries. I would suggest that many developed countries have different financial incentives in drug development than USA, and some of them may also find these approaches attractive.
2) The reader may get an unrealistic idea of potential efficacy of the proposed proactive approaches. I am sure the authors are aware of the potential difficulties in achieving good outcomes in disease like TN-IBC, which has deep and wide intrinsic resistance. It would be good to share with the reader what the potential limitations may be and possible ways of overcoming those obstacles. A commentary format offers an opportunity to propose imaginative solutions to difficult problems, without having to substantiate with data.
3) In this era of immunotherapy, it is easy to understand why everybody is focused on immunotherapy options. This reviewer feels that we need to simultaneously think about additional proactive therapies for overcoming intrinsic resistance along with immunotherapy. Just as the authors propose proactive safe, affordable, and effective approach to immunotherapy, they may also like to expand the concept to include additional therapies that are safe, affordable, and effective. Although it may be beyond the scope of the focused commentary, this may be worth a brief mention.
4) Since the commentary centers around a promising case report, many of those interested in this commentary would find it useful to know about the latest status of the patient along with any meaningful information resulting from the follow up. Therefore, it would be useful to update the commentary as needed. Usually journals are not interested in post-publication editing, but I hope exceptions can be made in cases such as this.
Author Response
Reviewer 1:
In this commentary, Alonso-Miguel et al. described proactive immunotherapy approaches for improving outcomes in inflammatory breast cancer, which is relative rare but very aggressive subtype. They did a good job in describing the limitations of current approaches in dealing with aggressive breast cancers such as IBC. They discussed the issues pertaining to clinical trials in sufficient detail and provided a good rationale for proactive immunotherapies that would be effective and financially affordable. Overall, the commentary is well written and suitable for publication in Cells.
Author response: We appreciate the positive evaluation of our commentary by this reviewer.
I have a few minor suggestions for the authors to consider.
- Authors suggested that based on the criteria of potential efficacy and low cost, some of the proposed therapies may be more suitable for clinical trials in poor countries. I would suggest that many developed countries have different financial incentives in drug development than USA, and some of them may also find these approaches attractive.
Author response: While we agree with this suggestion, our commentary is more focused on deploying FDA approved therapies that have demonstrated good results in other tumor types. As pointed out in the commentary, the cost of immunotherapies is almost prohibitive in poor countries and with approved or clinically tested drugs, there is no need for specific trials in IBC patients who are in urgent need and no time for long clinical trials.
- The reader may get an unrealistic idea of potential efficacy of the proposed proactive approaches. I am sure the authors are aware of the potential difficulties in achieving good outcomes in disease like TN-IBC, which has deep and wide intrinsic resistance. It would be good to share with the reader what the potential limitations may be and possible ways of overcoming those obstacles. A commentary format offers an opportunity to propose imaginative solutions to difficult problems, without having to substantiate with data.
Author response: This is a very helpful suggestion, and we appreciate it. See response to next query where we pooled our answer to reviewer’s points 2 and 3.
3) In this era of immunotherapy, it is easy to understand why everybody is focused on immunotherapy options. This reviewer feels that we need to simultaneously think about additional proactive therapies for overcoming intrinsic resistance along with immunotherapy. Just as the authors propose proactive safe, affordable, and effective approach to immunotherapy, they may also like to expand the concept to include additional therapies that are safe, affordable, and effective. Although it may be beyond the scope of the focused commentary, this may be worth a brief mention.
Author response: We appreciate suggestions 2 and 3 and accommodate the reviewer’s suggestions as indicated below. Aware of word limitations, we are trying to keep our response short. The following paragraph was added after line 351 (manuscript version with tracking):
- Treating a heterogeneous disease with complex combination therapies: Although ISV immunotherapies described above have demonstrated striking clinical responses, we should remember that, as observed in current ICIs approved to treat many cancers, these agents do not benefit all patients. Relapses are frequent because redundant pathways of immune suppression are active, important pathways of immune activation are silent, and critical tumor signaling pathways remain intact (Torres et al. 22). Hence, to overcome resistance and extend the benefit of immunotherapies to more patients, we need to precisely modulate various immunoregulatory and tumor intrinsic pathways. There are a series of safe and affordable agents that are approved or in clinical trials demonstrating good efficacy, including known chemotherapy drugs (doxorubicin, paclitaxel, cyclophosphamide), small inhibitor molecules (AKT inhibitors, MEK inhibitors, and PARP inhibitors), and TLR agonists (SD-101, an agonist for TLR9). Some of these agents are deployed as dual therapies in various cancers. The details of clinical trials in BC of these agents can be found in recent detailed reviews (Torres 22; Huppert 22, Nanda 22 and Schmid 22). Further, the ISV approach requires small quantities injected into the tumor parenchyma, reducing cost and potentially toxic events as well as offering the unique opportunity to combine multiple therapeutic agents to achieve a more robust immune response, higher objective response rates, and increased response duration in cancer patients, including IBC patients. While not the focus of this commentary, overcoming intrinsic resistance to therapies other than immunotherapy also requires ongoing and consistent consideration of new rational drug combinations that would be safe, affordable, and effective.
In addition, the text in the conclusions was edited as follows:
Line 377: There is sufficient data to support the application of ISV in IBC patients using the current approved immune checkpoint inhibitors alone or in combination with standard of care in the neoadjuvant setting to take advantage of a chemo naïve immune system not yet affected by toxicities from established chemotherapeutic regimens. Adding new and potent immunomodulators like CPMV, cavrotolimod, CFA, IL-2/IL-12, and other safe, affordable, and potentially effective therapies against this extremely aggressive and metastatic disease provides a pathway to significantly improve patients’ outcomes and eliminate IBC as a therapeutically orphan disease.
- Since the commentary centers around a promising case report, many of those interested in this commentary would find it useful to know about the latest status of the patient along with any meaningful information resulting from the follow up. Therefore, it would be useful to update the commentary as needed. Usually journals are not interested in post-publication editing, but I hope exceptions can be made in cases such as this.
Author response: We agree with this clinically relevant point. To accommodate this request, the following text was added:
Line 95: “We expect that this TN IBC patient will have a better outcome due to the treatment as compared with historical cases. This patient started chemo- and immunotherapy in August 2020 and underwent surgery in March 2021, followed by maintenance immunotherapy that was completed December 2021. As of this date, the patient is still clinically well and has no signs of recurrence (Dr. Dhakal’s personal communication).”
Reviewer 2 Report
The authors review a case report in which a TN inflammatory breast cancer was treated with chemotherapy and a checkpoint inhibitor, and then use this information to emphasize the need for new immunotherapeutic treatment approaches to these tumors. An important imitation of the review is that, while many statements are made indicating outcomes of clinical studies, rarely is any description given of the details of these studies – trial design, patient population, details of drug regimens, method of analysis, etc. A critical assessment of the trials should also be provided.
A Material and Methods section should be included to indicate how the literature search was conducted.
The immunotherapy concepts supporting these approaches should be described in detail.
The concept of intratumor injection of vaccine is proposed for these tumors. While the animal studies which are referenced employ tumors that are millimeters in size, it is difficult to image this approach being applicable to large, necrotic inflammatory breast cancers, especially ones in which the extent of neoantigens are low to begin with.
The limitations of the review should be discussed.
Author Response
Reviewer 2:
The authors review a case report in which a TN inflammatory breast cancer was treated with chemotherapy and a checkpoint inhibitor, and then use this information to emphasize the need for new immunotherapeutic treatment approaches to these tumors. An important imitation of the review is that, while many statements are made indicating outcomes of clinical studies, rarely is any description given of the details of these studies – trial design, patient population, details of drug regimens, method of analysis, etc. A critical assessment of the trials should also be provided.
Author response: We appreciate the suggestion and have considered it carefully. The commentary uses a case report with an impressive clinical response to propose thought-provoking ideas on how to apply currently approved immunotherapies like the one deployed in this TN IBC patient and/or immunotherapies in clinical trials with published striking clinical responses in both human and canine cancer patients. As a commentary, there is no standard format, and the intention is not to provide an extensive and detailed review of those studies as it would be in a review. When appropriate, we stated the existence of detailed reviews and provided the references.
A Material and Methods section should be included to indicate how the literature search was conducted.
Author response: There was not an organized attempt to comprehensively identify and note all relevant literature as would be done in a review where the search strategy is relevant. As a commentary rather than a review, the goal was to highlight some specific studies that support the concepts on which the commentary is focused. Therefore, there is no comprehensive search strategy to present to readers.
The immunotherapy concepts supporting these approaches should be described in detail.
Author response: As indicated above, the idea of the commentary is to highlight relevant clinical findings without duplicating the report of a given study. The reference of the original work is provided for those who want to look at the details about the rationale and other issues related to the presented study.
The concept of intratumor injection of vaccine is proposed for these tumors. While the animal studies which are referenced employ tumors that are millimeters in size, it is difficult to image this approach being applicable to large, necrotic inflammatory breast cancers, especially ones in which the extent of neoantigens are low to begin with.
Author response: We appreciate this observation because it highlights communication weaknesses. The tumors treated in our study and the studies we briefly highlighted in our commentary were not small murine tumors but rather large tumors, not in the range of millimeters but centimeters, and in a wide range of human tumors and canine tumors.
To clarify for readers and accommodate reviewer’s observation, the following changes were made in various sections. The denomination ‘large’ is due to the lack of specific information in the cited studies. However, it should be noted that within clinical trials, injected tumor sites should measure ≥1 cm in diameter (≥1.5 cm for lymph nodes) to ensure injectability.
Line 206: “…were also of the TN tumor subtype. These were large tumors (largest diameter of treated tumors ranged from 4 cm to 20 cm).
Line 277: “…in combination with intravenous pembrolizumab in patients with large advanced solid tumors, including melanomas, Merkel cell carcinomas, cutaneous squamous cell carcinomas, head and neck squamous cell carcinoma, leiomyosarcoma, and various metastatic tumor patients [50].”
Line 290: “…. in three preclinical species (relatively large mouse mastocytomas, canine mastocytomas, and equine melanomas) and in large human tumors (one non-small cell lung cancer, one metastatic osteosarcoma, one squamous cell cervical carcinoma, one squamous cell head and neck cancer, one prostate cancer, two metastatic renal cancers, two lung cancers, one urothelial cancer, and two invasive hormone receptor positive ductal breast cancers).
Line 335: This smart design demonstrated high activity and tolerability in large canine soft tissue sarcoma (STS) and oral melanoma (OM) patients (52).
Reviewer: “…it is difficult to image this approach being applicable to large, necrotic inflammatory breast cancers, especially ones in which the extent of neoantigens are low to begin with”,
Author response: We provided a summary of our paper highlighting the striking positive response observed in our canine inflammatory mammary cancer (IMC) patients with large tumors. The data support the further deployment of our approach in human TN IBC (lines 177-190 and 199-211).
Further, the issue of neoantigens is of interest as they may play an important role in response to immunotherapies. However, as we mentioned in line 205, we have also observed striking clinical responses in luminal A and luminal B canine IMC patients. These are considered ‘cold tumors’ in human breast cancer patients. In addition, to highlight the efficacy of immunomodulators in other non-TN breast tumors, we edited this text discussing the Fahrer’s study (line 294) ‘two invasive ductal breast cancers’ now reads ‘two invasive hormone receptor positive ductal breast cancers.’
Reviewer: The limitations of the review should be discussed.
Author response: As discussed above, we specifically drafted a commentary and never intended to generate a comprehensive review. The goal was to highlight some limited but significant new data in the field and propose proactive solutions to solve serious problems as in the case of IBC.
Reviewer 3 Report
The commentary, “Proactive immunotherapeutic approaches against inflammatory breast cancer will improve patient outcomes” (cells-1810896) by Alonso-Miguel et al., provides novel therapeutic potential of an immune checkpoint inhibitor in combination with anthracycline-taxane-based chemotherapy to treat TN IBC. Since TN IBC is very rare disease, a lack of clinical trials, at least in part, causes the lack of effective therapies against TN IBC. They point out that ISV with reduced amount of immune checkpoint inhibitors is a potential treatment option, instead of high dose immune checkpoint inhibitor therapy, for IBC patients with reduced economic burden. The manuscript well summarizes the recent approaches and provide insights how the cost of treatment could be reduced by applying more affordable immunotherapeutics. Before processing further, a minor revision is suggested as follows:
Minor comments
Lines 197 – 198: Meaning of this sentence is not clear. Please, modify this.
Author Response
Reviewer 3:
The commentary, “Proactive immunotherapeutic approaches against inflammatory breast cancer will improve patient outcomes” (cells-1810896) by Alonso-Miguel et al., provides novel therapeutic potential of an immune checkpoint inhibitor in combination with anthracycline-taxane-based chemotherapy to treat TN IBC. Since TN IBC is very rare disease, a lack of clinical trials, at least in part, causes the lack of effective therapies against TN IBC. They point out that ISV with reduced amount of immune checkpoint inhibitors is a potential treatment option, instead of high dose immune checkpoint inhibitor therapy, for IBC patients with reduced economic burden. The manuscript well summarizes the recent approaches and provide insights how the cost of treatment could be reduced by applying more affordable immunotherapeutics. Before processing further, a minor revision is suggested as follows:
Author response: We appreciate the reviewer’s positive evaluation of our commentary.
Minor comments
Lines 197 – 198: Meaning of this sentence is not clear. Please, modify this.
Author response: We appreciate this observation. The sentence has been edited as follows (lines 200-203): CPMV nanoparticles are agonists for toll-like receptors (TLR) 2, 4, and 7 with TLR 2,4 recognizing the viral capsid and TLR7 recognizing the viral RNA genome. Because eCPMV lacks RNA, it exerts activity only through TLR2 and TLR4 [44].
Round 2
Reviewer 1 Report
The authors have addressed the minor points raised in the initial review.
Author Response
Reviewer 1:
In this commentary, Alonso-Miguel et al. described proactive immunotherapy approaches for improving outcomes in inflammatory breast cancer, which is relative rare but very aggressive subtype. They did a good job in describing the limitations of current approaches in dealing with aggressive breast cancers such as IBC. They discussed the issues pertaining to clinical trials in sufficient detail and provided a good rationale for proactive immunotherapies that would be effective and financially affordable. Overall, the commentary is well written and suitable for publication in Cells.
Author response: We appreciate the positive evaluation of our commentary by this reviewer.
I have a few minor suggestions for the authors to consider.
- Authors suggested that based on the criteria of potential efficacy and low cost, some of the proposed therapies may be more suitable for clinical trials in poor countries. I would suggest that many developed countries have different financial incentives in drug development than USA, and some of them may also find these approaches attractive.
Author response: While we agree with this suggestion, our commentary is more focused on deploying FDA approved therapies which have demonstrated good results in other tumor types. As pointed out in the commentary, the cost of immunotherapies is almost prohibitive in poor countries and with approved or clinically tested drugs, there is no need for specific trials in IBC patients who are in urgent need and no time for long clinical trials.
- The reader may get an unrealistic idea of potential efficacy of the proposed proactive approaches. I am sure the authors are aware of the potential difficulties in achieving good outcomes in disease like TN-IBC, which has deep and wide intrinsic resistance. It would be good to share with the reader what the potential limitations may be and possible ways of overcoming those obstacles. A commentary format offers an opportunity to propose imaginative solutions to difficult problems, without having to substantiate with data.
Author response: This is a very helpful suggestion and we appreciate it. See response to next query where we pooled our answer to reviewer’s points 2 and 3.
3) In this era of immunotherapy, it is easy to understand why everybody is focused on immunotherapy options. This reviewer feels that we need to simultaneously think about additional proactive therapies for overcoming intrinsic resistance along with immunotherapy. Just as the authors propose proactive safe, affordable, and effective approach to immunotherapy, they may also like to expand the concept to include additional therapies that are safe, affordable, and effective. Although it may be beyond the scope of the focused commentary, this may be worth a brief mention.
Author response: We appreciate suggestions 2 and 3 and accommodate the reviewer’s suggestions as indicated below trying to keep it short due to space limitations. The following paragraph was added after line 344:
- Treating a heterogeneous disease with complex combination therapies: Although ISV immunotherapies described above have demonstrated striking clinical responses, we should remember that, as observed in currently ICIs approved to treat many cancers, these agents do not benefit all patients. Relapses are frequent because redundant pathways of immune suppression are active, important pathways of immune activation are silent, and critical tumor signaling pathways remained intact (Torres et al. 22). Hence, to overcome resistance and to extend the benefit of immunotherapies to more patients, we need to precisely modulate various immunoregulatory and tumor intrinsic pathways. There are a series of safe and affordable agents that are approved or in clinical trials demonstrating good efficacy, including known chemotherapy drugs (doxorubicin, paclitaxel, cyclophosphamide), small inhibitor molecules (AKT inhibitors, MEK inhibitors, and PARP inhibitors), and TLR agonists (SD-101, an agonist for TLR9). Some of these agents are being deployed as dual therapies in various cancers. The details of clinical trials in BC of these agents can be found in recent detailed reviews (Torres 22; Huppert 22, Nanda 22 and Schmid 22). Further, the ISV approach requires small quantities injected into the tumor parenchyma, reducing cost and potentially toxic events as well as offering the unique opportunity to combine multiple therapeutic agents to achieve a more robust immune response, higher objective response rates, and increased response duration in cancer patients, including IBC patients. While not the focus of this commentary, overcoming intrinsic resistance to therapies other than immunotherapy also requires ongoing and consistent consideration of new rational drug combinations that would be safe, affordable and effective.
In addition, the text in the conclusions was also edited as follows:
Line 348: There is sufficient data to support the application of ISV in IBC patients using the current approved immune checkpoint inhibitors alone or in combination with standard of care in the neoadjuvant setting to take advantage of a chemo naïve immune system not yet affected by toxicities from established chemotherapeutic regimens. Adding new and potent immunomodulators like CPMV, cavrotolimod, CFA, IL-2/IL-12, and other safe, affordable, and potentially effective therapies against this extremely aggressive and metastatic disease provides a pathway to significantly improve patients’ outcomes and eliminate IBC as a therapeutically orphan disease.
- Since the commentary centers around a promising case report, many of those interested in this commentary would find it useful to know about the latest status of the patient along with any meaningful information resulting from the follow up. Therefore, it would be useful to update the commentary as needed. Usually journals are not interested in post-publication editing, but I hope exceptions can be made in cases such as this.
Author response: We agree with this clinically relevant point. To accommodate this request, the following text was added:
Line 95: “It is expected, therefore, that this TN IBC patient will have a better outcome due to the treatment and as compared with historical cases. This patient started chemo- and immunotherapy in August 2020 and underwent surgery in March 2021, followed by maintenance immunotherapy which was completed December 2021. As of this date, the patient is still clinically well and has no signs of recurrence (Dr. Dhakal’s personal communication).”
Reviewer 2 Report
The authors have addressed the concern about tumor size in the management plan, but have not addressed any of the other concerns or questions.
Author Response
Reviewer 2:
The authors review a case report in which a TN inflammatory breast cancer was treated with chemotherapy and a checkpoint inhibitor, and then use this information to emphasize the need for new immunotherapeutic treatment approaches to these tumors. An important imitation of the review is that, while many statements are made indicating outcomes of clinical studies, rarely is any description given of the details of these studies – trial design, patient population, details of drug regimens, method of analysis, etc. A critical assessment of the trials should also be provided.
Author response: We appreciate the suggestion and have considered it carefully. The commentary uses a case report with an impressive clinical response to propose thought-provoking ideas on how to apply currently approved immunotherapies like the one deployed in this TN IBC patient and/or immunotherapies in clinical trials with published striking clinical responses in both human and canine cancer patients. As a commentary, there is no standard format, and the intention is not to provide an extensive and detailed review of those studies as it would be in a review. When appropriate, we stated the existence of detailed reviews and provided the references.
A Material and Methods section should be included to indicate how the literature search was conducted.
Author response: There was not an organized attempt to comprehensively identify and note all relevant literature as would be done in a review where the search strategy is relevant. As a commentary rather than a review, the goal was to highlight some specific studies that support the concepts on which the commentary is focused. Therefore, there is no comprehensive search strategy to present to readers.
The immunotherapy concepts supporting these approaches should be described in detail.
Author response: As indicated above, the idea of the commentary is to highlight relevant clinical findings without duplicating the report of a given study. The reference of the original work is provided for those who want to look at the details about the rationale and other issues related to the presented study.
The concept of intratumor injection of vaccine is proposed for these tumors. While the animal studies which are referenced employ tumors that are millimeters in size, it is difficult to image this approach being applicable to large, necrotic inflammatory breast cancers, especially ones in which the extent of neoantigens are low to begin with.
Author response: We appreciate this observation because it highlights communication weaknesses. The tumors treated in our study and the studies we briefly highlighted in our commentary were not small murine tumors but rather large tumors, not in the range of millimeters but centimeters, and in a wide range of human tumors and canine tumors.
To clarify for readers and accommodate reviewer’s observation, the following changes were made in various sections. The denomination large is due to the lack specific information in the cited studies. However, it should be noted that within clinical trials, injected tumor sites should measure ≥1 cm in diameter (≥1.5 cm for lymph nodes) to ensure injectability.
Line 201: “…were also of the TN tumor subtype. These were large tumors (largest diameter of treated tumors ranged from 4 cm to 20 cm).
Line 270: “…in combination with intravenous pembrolizumab in patients with large advanced solid tumors, including melanomas, Merkel cell carcinomas, cutaneous squamous cell carcinomas, head and neck squamous cell carcinoma, leiomyosarcoma, and various metastatic tumor patients [50].”
Line 2817: “…. in three preclinical species (relatively large mouse mastocytomas, canine mastocytomas, and equine melanomas) and in large human tumors (one non-small cell lung cancer, one metastatic osteosarcoma, one squamous cell cervical carcinoma, one squamous cell head and neck cancer, one prostate cancer, two metastatic renal cancers, two lung cancers, one urothelial cancer, and two invasive hormone receptor positive ductal breast cancers).
Line 327: This smart design demonstrated high activity and tolerability in large canine soft tissue sarcoma (STS) and oral melanoma (OM) patients (52).
Reviewer: “…it is difficult to image this approach being applicable to large, necrotic inflammatory breast cancers, especially ones in which the extent of neoantigens are low to begin with”,
Author response: We provided a summary of our paper highlighting the striking positive response observed in our canine inflammatory mammary cancer (IMC) patients with large tumors. The data support the further deployment of our approach in human TN IBC (lines 177-190 and 199-211).
Further, the issue of neoantigens is of interest and they may play an important role in response to immunotherapies. However, as we mentioned in line 205, we have also observed striking clinical responses in luminal A and luminal B canine IMC patients. These are considered ‘cold tumors’ in human breast cancer patients. In addition, to highlight the efficacy of immunomodulators in other non-TN breast tumors, we change this text discussing the Fahrer’s study (line 287) ‘two invasive ductal breast cancers’ now reads ‘two invasive hormone receptor positive ductal breast cancers.’
Reviewer: The limitations of the review should be discussed.
Author response: As discussed above, we specifically drafted a commentary and never intended to generate a comprehensive review. The goal was to highlight some limited but significant new data in the field and propose proactive solutions to solve serious problems as in the case of IBC.
Reviewer 3 Report
Thank you for your kind revision.
Author Response
The reviewer accepted our responses.